

# COVID-19: Taiwan's epidemiological characteristics and public and hospital responses

Chih-Ming Chang[1,2], Ting-Wan Tan[1], Tai-Cheng Ho[1], Chung-Chu Chen[3,4], Tsung-Hsien Su[5,6] and Chien-Yu Lin[6,7]

[1] Department of Nursing, Hsinchu MacKay Memorial Hospital, Hsinchu, Taiwan
[2] Department of Healthcare Management, Yuanpei University of Medical Technology, Hsinchu, Taiwan
[3] Department of Internal Medicine, Hsinchu MacKay Memorial Hospital, Hsinchu, Taiwan
[4] Teaching Center of Natural Science, Minghsin University of Science and Technology, Hsinchu, Taiwan
[5] Department of Obstetrics and Gynecology, Hsinchu MacKay Memorial Hospital, Hsinchu, Taiwan
[6] Department of Medicine, MacKay Medical College, New Taipei, Taiwan
[7] Department of Pediatrics, Hsinchu MacKay Memorial Hospital, Hsinchu, Taiwan

Corresponding author
Chien-Yu Lin,
mmhped.lin@gmail.com

## ABSTRACT

**Background:** Coronavirus disease 19 (COVID-19) is a global health threat with significant medical, economic, social and political implications. The optimal strategies for combating COVID-19 have not been fully determined and vary across countries.
**Methods:** By the end of February 2020 in Taiwan, 2,150 patients received diagnostic COVID-19 testing and 39 confirmed cases were detected. This is a relatively lower rate of infection compared to other Asian countries. In this article, we summarize the epidemiological characteristics of the 39 infected patients as well as public and hospital responses to COVID-19.
**Results:** Thirty-nine COVID-19 cases and one death have been confirmed in Taiwan. Seventeen of these patients were infected by family members or in hospital wards, emphasizing how COVID-19 is mostly spread by close contact. We examined how hospital have responded to COVID-19, including their implementation of patient route control, outdoor clinics, hospital visit restrictions and ward and staff modifications. We also studied the public's use of face masks in response to COVID-19. These strategies may reduce the spread of COVID-19 in other countries.
**Conclusion:** The emergence and spread of COVID-19 is a threat to health worldwide. Taiwan has reported lower infected cases and its strategies may contribute to further disease prevention and control.

# INTRODUCTION

The novel coronavirus, 2019-nCoV, emerged in Wuhan, China in December 2019 and has rapidly spread across the world (*Gates, 2020*; *Zhu et al., 2020*). In February 2020, the World Health Organization (WHO) renamed this epidemic disease coronavirus disease

(COVID-19) as it is caused by severe acute respiratory syndrome coronavirus 2 (SARS-CoV-2). The number of COVID-19 cases has rapidly increased worldwide (*WHO, 2020*; *Del Rio & Malani, 2020*). As of 29 February 2020, there were 85,403 confirmed COVID-19 cases and 2,924 fatalities across 49 countries (*WHO, 2020*; *Dong, Du & Gardner, 2020*). Although the emerging threat of COVID-19 has drawn global attention, the optimal strategies to reduce the spread of disease remain largely undetermined (*Day, 2020*).

Several studies have investigated the virology, transmission, risk factors and protection associated with COVID-19 (*Chen et al., 2020a*; *Guan et al., 2020*; *Huang et al., 2020*; *Wang et al., 2020*; *Wu & McGoogan, 2020*; *Yang et al., 2020*). COVID-19 spreads by human-to-human transmission, primarily through respiratory droplets and direct contact with groups of infected family members, friends, colleagues, or medical health workers. However, even asymptomatic patients can be a source of infection. COVID-19 has dramatically higher rates of transmission compared to SARS and Middle East respiratory syndrome, but an observed lower fatality rate. The median age of COVID-19 patients ranged from 47 to 56 years old; approximately 65% of patients were male and 35% were female. Approximately 23.7% of adult COVID-19 patients have at least one of the following coexisting underlying chronic illnesses: cardiovascular disease, diabetes mellitus, hypertension, or chronic obstructive lung disease. The most common reported symptoms are fever (43.8% on admission and 88.7% during hospitalization), dry cough (67.8%), dyspnea (18.7%), myalgia (14.9%), headache (13.6%), diarrhea (3.8%), sore throat (13.9%), nasal obstruction (4.8%) and fatigue (38.1%). The optimal infectious control measurements for COVID-19 have not yet been established (*Razai et al., 2020*).

COVID-19 is continuing to dramatically spread across many countries (*WHO, 2020*). However, Taiwan has a relatively lower rate of infected cases. After the first diagnosed case in Taiwan on 21 January 2020, 39 cases were confirmed as of 29 February 2020. Some cases were imported by travelers and others were within families. We conducted this study to summarize the epidemiological characteristics of the 39 cases and to propose technological advances and infectious control measurements that may contribute to reducing disease spread (*Chen et al., 2020b*; *Wang, Ng & Brook, 2020*). We also discuss some of the Taiwanese public health's and hospital's responses to COVID-19.

## MATERIALS AND METHODS

### Study design and data sources

We conducted this retrospective observational study at the end of February 2020 to investigate the epidemiological characteristics of the first 39 confirmed COVID-19 cases in Taiwan. The Taiwan Centers for Disease Control (TCDC) enforced many policies to combat the spread of COVID-19 (*Taiwan Centers for Disease Control, 2020*). The epidemiological data of the confirmed cases were released to the public. Although the identifiable data were encrypted, the clinical presentations and epidemiological relationships were reported, and data regarding demographics, clinical symptoms, contact

history, and onset dates were extracted. We summarized the public data to show the clinical characteristics of the 39 cases.

## COVID-19 diagnoses

Since COVID-19 is considered a communicable disease in Taiwan, all suspected cases are required to be reported to the TCDC. The clinical COVID-19 diagnostic criteria include a history of travel or residence in endemic areas, contact with confirmed cases, contact with cases of fever or respiratory symptoms, radiological image characteristics of pneumonia, and clinical manifestations of fever or respiratory tract symptoms. Early diagnosis and timely containment are important. The TCDC's diagnostic testing criterion included: (1) patients from endemic areas with respiratory symptoms; (2) patients with unexplained pneumonia or community pneumonia and poor responses to antibiotic treatment; (3) patients in close contact with confirmed cases; and (4) cluster infections or unwell people in specific workplaces, such as hospitals or nurseries. The defined endemic areas varied over time.

Suspected cases were confirmed by nucleic acid detection from the sputum, throat swab, lower respiratory tract secretion, or blood samples. The polymerase chain reaction for COVID-19 and other viruses were performed (*Cui, Li & Shi, 2019*; *Lin et al., 2020*; *Lu et al., 2020*; *The Lancet, 2020*). The suspected cases were isolated in negative pressure rooms or single bedrooms until a second negative test result.

## COVID-19 treatment

Since there is no specific treatment for COVID-19, the main treatments used are supportive care and oxygen support (*WHO, 2020*; *Taiwan Centers for Disease Control, 2020*). Suspected or confirmed cases were moved to isolated rooms and received supportive treatment to ensure electrolyte balance, closely monitored vital signs, and oxygen saturation. Routine blood tests and arterial blood gas analyses were administered if necessary. Chest imaging and other procedures were arranged. Patients with pneumonia received intravenous broad-spectrum antibiotics such as penicillin or ceftriaxone, and oseltamivir was also administered empirically. For confirmed cases, they could leave isolation and return home after three consecutive negative PCR tests. In average, it took approximately 1 month to achieve the criteria of de-isolation thus the capacity of isolated rooms was an important issue.

## Public health and hospital responses

Widespread anxiety is anticipated when facing a novel and unknown infection (*The Lancet, 2020*). The TCDC executed several policies, but individual hospitals were authorized to implement their own infective control measurements against COVID-19 in addition to their normal practices. For example, decreasing exposure was important to reduce disease spread, and hospitalized patients are susceptible to various kinds of infectious diseases including COVID-19. To reduce viral transmission, hospitals set up outdoor clinics to care for patients with travel or contact histories. We investigated other important health polices and hospital responses.

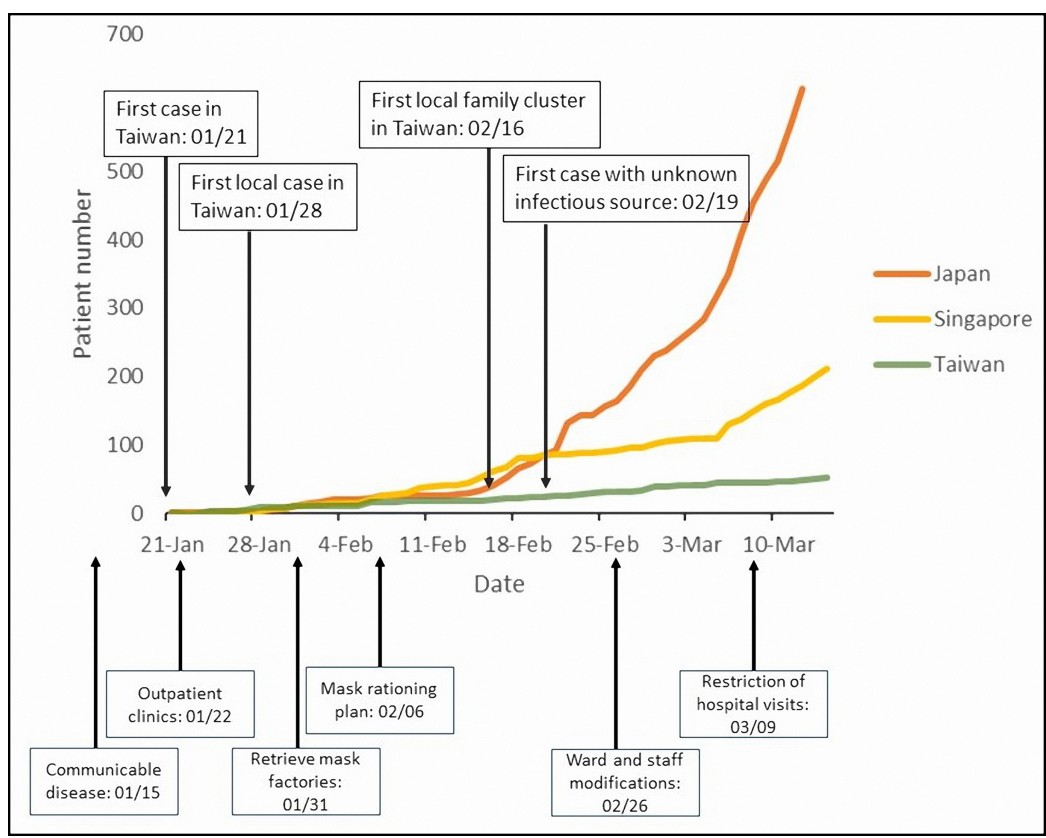

**Figure 1 Cumulative cases of COVID-19 in some Asian countries since middle February.**

## RESULTS

### Epidemiological characteristics of confirmed cases

We summarized cumulative cases from other Asian countries starting in mid-February (Fig. 1). By the end of February 2020, 2,150 patients in Taiwan had been tested for COVID-19. Among them, 39 (1.8%) patients tested positive for SARS-CoV-2.
The epidemiological characteristics of the 39 confirmed cases are summarized in Table 1. Sixteen (41.0%) patients were male and 23 (59.0%) were female. Approximately half (53.8%) of these patients were between 41 and 60 years old. Nineteen cases (48.7%) had a history of overseas travel or residence or had visited China. Fifteen (38.5%) of the confirmed cases had direct contact transmission from infected family members.
One patient (2.6%) died. The TCDC aggressively tracked the travel, occupation, contact, and cluster histories of the confirmed cases. The correlations among the 39 cases were plotted and publicly released (Fig. 2).

### Public health and hospital responses to COVID-19
#### Home quarantine and patient route control

Hospitals designated separate entrances and exits for patients and the public to prevent hospital-acquired COVID-19 infections. Suspected cases were treated separately from the

**Table 1 Epidemiological characteristics of 39 confirmed cases.**

| Clinical characteristics | | No. | Percentage (%) |
|---|---|---|---|
| Total | | 39 | 100 |
| Gender | Male | 21 | 53.8 |
| | Female | 18 | 46.2 |
| Age (years old) | 11~20 | 2 | 5.1 |
| | 21~30 | 4 | 10.3 |
| | 31~40 | 4 | 10.3 |
| | 41~50 | 10 | 25.6 |
| | 51~60 | 11 | 28.2 |
| | 61~70 | 3 | 7.7 |
| | 71~80 | 3 | 7.7 |
| | >81 | 2 | 5.1 |
| Travel history | None | 20 | 51.3 |
| | Yes | 19 | 48.7 |
| Family cluster infection | None | 21 | 53.8 |
| | Yes | 17 | 43.6 |
| | Unknown source | 1 | 2.6 |

emergency and outpatient cases throughout the duration of their hospital stay. When scheduling appointments online, the Tracking and Management Mechanism for People under Infection Risk announcement appeared (Fig. 3). If patients had any history of travel to endemic areas or of respiratory infection, they were guided to a special COVID-19 outdoor clinic (Fig. 4).

Suspected cases were guided to an outdoor triage and then admitted to an isolation room. Diagnostic tests were performed in negative pressure rooms or single bedrooms. Hand sanitizer, masks and COVID-19-related posters were posted at the hospital entrances. Patients and visitors had to wash their hands with alcohol hand gel before entering the hospital. These infectious control measurements all contributed to reducing hospital-acquired infections.

### Hospital visit restrictions

Hospital visits are common in Asian cultures, so hospitals in Taiwan restricted visits to prevent COVID-19 infections and outbreaks. Instead of face-to-face hospital visits, video calls were recommended for family and friends to contact patients. Ward entrances were controlled by electromagnetic doors requiring access cards. Patients were not allowed in restricted areas. No more than two visitors were allowed to visit the same patient at the same time. Visitors with recent travel histories were restricted from entering the hospital (Fig. 5). In the case of a confirmed infection, visitor registration provided clues for tracking the infectious source and their contacts.

### Regulation of masks and other personal protective equipment

Surgical masks are basic protections that prevent human-to-human transmission of COVID-19. Taiwanese people tend to wear masks in large and crowded gathering areas in

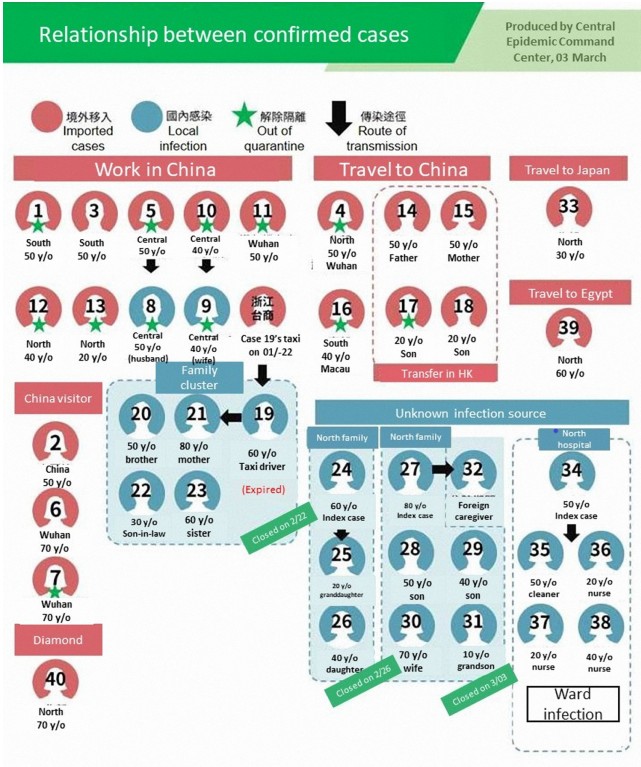

**Figure 2 The relationship and route of transmission of 39 confirmed cases in Taiwan.**

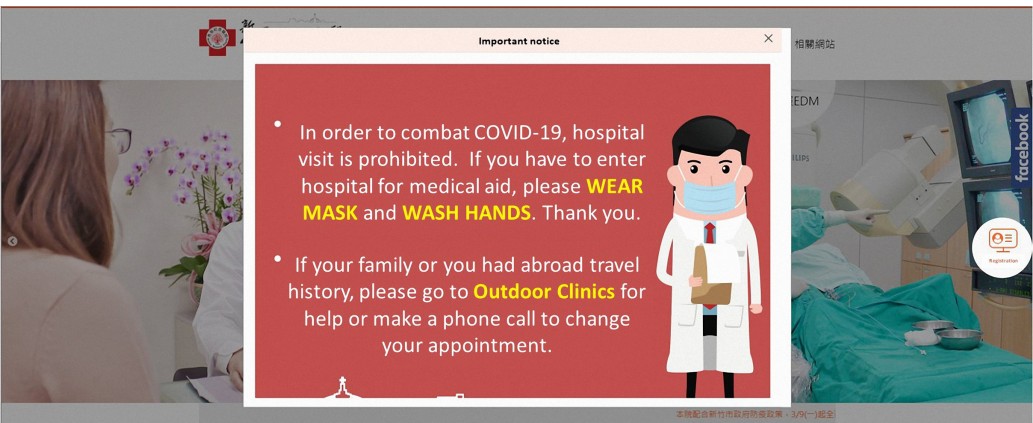

**Figure 3 The pop-up window before internet registration to remind triage with travel history or respiratory symptoms.** If medical aid for people with abroad travel history, they will be guided to a specific outdoor clinic.

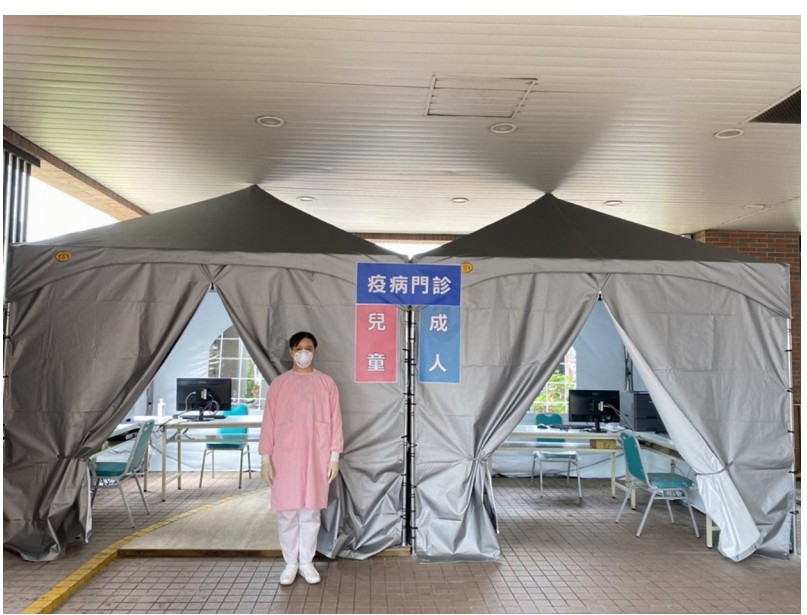

**Figure 4 The outdoor clinics for patients with travel history or respiratory symptoms.**

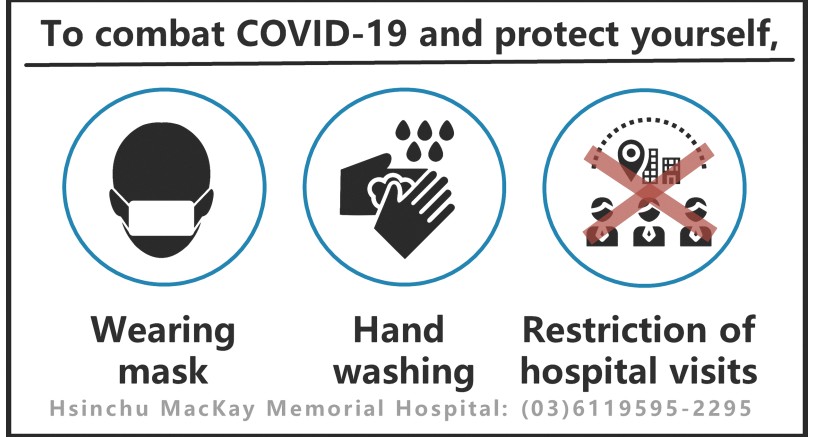

**Figure 5 A shorter version of visitor restriction was posted to provide clear instructions to patients and visitors.** If visitors were allowed to enter hospital, registration with detailed information was required for further contact tracing.

their daily lives to protect themselves and others. In order to prevent public over-purchase and hoarding, the Taiwanese government took actions to recruit the mask factories and allocate all masks. The government helped factories upgrade their facilities, and mask production increased from 1 million per day in early February to 10 million per day in late February. The Taiwanese government issued surgical masks to all local hospitals and clinics to prevent shortages. Each medical staff member received one to two masks every day for regular use, and additional masks or N95 masks were provided for healthcare providers taking care of COVID-19 patients or performing invasive procedures. Moreover, the government provided surgical masks for susceptible populations, such as patients with

cancer, hemodialysis, or those undergoing chemotherapy and radiotherapy. Upon hospital admission, one mask was given to patients with reduced nosocomial infection or hospital-acquired infection. To reduce community transmission, the government also issued surgical masks to children. Taiwanese residents needed to present a national health insurance card to purchase surgical masks from local pharmacies and primary healthcare units. This policy allowed each residence to purchase two masks weekly and prohibited repeat purchases.

### Ward and staff modifications

Suspected and confirmed cases were admitted to their own rooms to ensure isolation. In order to minimize infections of healthcare providers, nursing staff adhered to group scheduling and fixed shifting, separated by meal breaks. Job and ward rotations were prohibited to reduce the chances of hospital-acquired transmission.

## DISCUSSION

COVID-19 is a global health threat, but Taiwan has a relatively lower rate of infection compared to other countries. Approximately 50% of Taiwan's 39 confirmed cases were imported and 38.5% were infected by their own families. Aggressive tracking and epidemiological surveillance strategies contributed to identifying infectious sources and reducing disease spread. Furthermore, infectious control measurements in hospitals may have reduced the transmission of COVID-19.

There are no established best strategies against COVID-19 and policies differ across populations, geographic regions, cultures, countries and time. "Lockdown" was a powerful strategy used by China to isolate infected people and reduce transmission, but it is not easily replicated in many countries. The United Kingdom's "herd immunity" strategy also caught our attention (*Yong, 2020*). However, Taiwan's close proximity to and frequent interactions with China suggest that following the UK's strategy could result in the collapse of the medical system. Moreover, convenience in traveling contributed to virus transmission thus disease spread was almost inevitable. The rapid spread of COVID-19 had been observed in UK and the "herd immunity" strategy had been abandoned. The UK government made efforts towards "containment" to slow down the virus transmission (*Boyle, 2020*). We were still exploring the appropriate strategies to tackle COVID-19 in different situations.

Aggressive control measurements in Taiwan helped flatten the infectious curve, prevent medical collapse, and reduce case fatality rates. For example, there were 1,300 isolation rooms with an occupancy rate of approximately 70% in February 2020 (*Taiwan Centers for Disease Control, 2020*). Adequate isolation rooms ensured adequate containment and reduced the risk of community infections. Furthermore, wearing masks is already prevalent in Taiwanese society, and masks provide simple and direct protection against droplet infection (*Leung et al., 2020*). Wearing masks has not been a common practice in other countries (*Desai & Mehrotra, 2020*). Taiwan's aggressive strategies ensured the provision of adequate masks, decreasing both anxiety among the public and disease spread.

Although the culture and population density of Japan are relatively similar to Taiwan's, Japan's situation was more complicated. They faced the complex issues with the Diamond Princess cruise ship carrying infected patients, quarantined in the port of Yokohama, Japan, and the Olympics, and they used different strategies to combat COVID-19 (*Gallego et al., 2020*; *Montanari, 2020*). Optimal strategies varied across different countries and changed over time.

The COVID-19 pandemic is a rapidly changing situation. According to a study by the Chinese CDC, the majority of infected people are aged 30–79 years (87%). In Taiwan's 39 confirmed cases, 35 (90%) were aged 30–79 years. Taiwan's case fatality rate (1/39, 2.56%) was similar to China's (1,023/44,672, 2.3%) (*Wu & McGoogan, 2020*). Approximately half of Taiwanese patients (19/39, 48.7%) had an overseas travel history, but the high risk of community infection should not be ignored. Family and ward clusters are important transmission sources, and 22 people across five families were identified in the 39 reported cases. The high contagiousness of SARS-CoV-2 makes disease prevention difficult. Moreover, 81% of infected patients reported mild symptoms which were easily overlooked leading to delayed diagnoses (*Bai et al., 2020*; *Rothe et al., 2020*). Aggressive tracking and infectious source identification lead to early quarantine, transmission prevention, and early diagnoses of patients with mild symptoms.

Although SARS-CoV-2's complete route of transmission has not been completely identified, it is mainly spread via droplets and physical contact (*Li et al., 2020*; *Lu, Liu & Jia, 2020*; *Xu et al., 2020*). Wearing masks and washing hands are essential and effective in preventing the spread of infectious disease (*Leung, Lam & Cheng, 2020*; *MacIntyre & Chughtai, 2015*). Public panic is expected when an emerging disease appears likely to become endemic or a widespread epidemic (*Bao et al., 2020*; *Medley & Vassall, 2017*), leading to a shortage of masks and increased risk of disease transmission. The TCDC's mask policy is an executable and effective strategy to combat disease. Furthermore, hospitals implementing visitation restrictions and outdoor clinics may also reduce the risk of COVID-19 transmission.

Technological advances also contribute to improvements in infection control. Big data analytics, new technology, and proactive testing have been applied in the war against COVID-19 (*Wang, Ng & Brook, 2020*). A real-time, interactive internet dashboard has also been utilized to provide timely information for the general public and healthcare providers (*Dong, Du & Gardner, 2020*). These strategies are believed to be beneficial in preventing and controlling disease spread. We summarized some additional hospital responses and provided photographs for healthcare providers' reference (Figs. 1, 3, 4 and 5).

Our study had some limitations. First, COVID-19 is a communicable disease and universal screening is not currently available. The number of patients with mild symptoms may be underestimated. Second, since SARS-CoV-2's virology, transmission, incubation period, and contagious period are not fully understood, the most effective strategies to use against COVID-19 remain speculative. More studies are required to investigate the effectiveness of individual infectious control measurements.

## CONCLUSIONS

The emergence of COVID-19 is a critical global issue. Taiwan's 39 confirmed cases showed a similar age distribution as previous studies (*Guan et al., 2020*). The role of family and ward clusters in disease transmission was emphasized. Public health and hospitals' patient route control, outdoor clinics, hospital visit restrictions, mask regulations and ward and staff modification strategies may contribute to reducing the transmission of COVID-19.

## ACKNOWLEDGEMENTS

We thank the Centers for Disease Control, Taiwan and all efforts to combat COVID-19. We thank David Singleton for his brilliant review and constructive comments. The quality of our manuscript has been improved following his precious suggestions.

### Funding

The authors received no funding for this work.

### Competing Interests

The authors declare that they have no competing interests.

### Author Contributions

- Chih-Ming Chang conceived and designed the experiments, performed the experiments, analyzed the data, prepared figures and/or tables, authored or reviewed drafts of the paper, and approved the final draft.
- Ting-Wan Tan performed the experiments, analyzed the data, prepared figures and/or tables, and approved the final draft.
- Tai-Cheng Ho conceived and designed the experiments, analyzed the data, prepared figures and/or tables, and approved the final draft.
- Chung-Chu Chen performed the experiments, analyzed the data, authored or reviewed drafts of the paper, and approved the final draft.
- Tsung-Hsien Su analyzed the data, authored or reviewed drafts of the paper, and approved the final draft.
- Chien-Yu Lin conceived and designed the experiments, performed the experiments, analyzed the data, authored or reviewed drafts of the paper, and approved the final draft.

### Data Availability

The data are publicly available online at:

1. The United Daily News (2020). "Graphical illustration of coronavirus cases in Taiwan". Interactive dataset. https://udn.com/newmedia/2020/covid-19-taiwan/?fbclid=IwAR3Q6SFJ7F2qVj7XSxHKqHF4O9xBxrOv4IEx9u0JzIM3E_bSwnjDiP8Ojdc.

2. Feng (2020). "Area, Age, and Gender Statistical table—19CoV". Center for Diseases Control, Ministry of Health and Welfare. Dataset. https://data.cdc.gov.tw/en/dataset/agsdctable-day-19cov.

The most current data is available at the Taiwan CDC: https://nidss.cdc.gov.tw/en/SingleDisease.aspx?dc=1&dt=5&disease=19CoV.

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
