# Peer review of "COVID-19: Taiwan’s epidemiological characteristics and public and hospital responses"

_PeerJ, doi:10.7717/peerj.9360_

## Round 0.1 · original submission · Major Revisions

Please, analyze the reviewers' suggestions so as to improve your paper. It is important to publish information related to the COVID-19 as soon as possible.

Reviewer 1 ·

Basic reporting

The paper provides epidemiological characteristics of the 39 confirmed COVID-19 cases in Taiwan, one of the countries with lower rates of the disease. This is a very informative paper for other countries and needs to be published very fast.

Experimental design

The experimental design is a serial study case, a type of observational study in epidemiology. I think that is important to make this is clear in the manuscript methodology section. It is important to improve the methodology section and inform that it was used statistical descriptive analysis. It is necessary to inform what variables were collected in relation to the patients, such as, age, gender, etc. It is not clear. I know the variables because I read the results section, but this information need to stay in methods as well.

Validity of the findings

This is an important description of COVID-19 cases and health system responses in Taiwan. This is necessary to help and compare with other countries. All of this information is essential.

Additional comments

Line 60 and 61: format the reference.
Line 61: format the date.
Check the numbers formatting according to the English language.
Line 119: change 2019-nCoV to COVID-19.
Line 147: format the percentage.
Table 1: check the formatting and the uppercase of the words.
Figure 3 and figure 5: important to have in the supplementary files or together with the figure, as a panel figure, the traduction to English.
Figure 5, try to improve the pictures, they are distorted.
Line 193, 195, 198, 200 and 203: fix the term facemask.

·

Basic reporting

Thank you for giving me the opportunity to review this interesting and potentially important manuscript. I found the manuscript to be practically focused on potentially effective control measures to contain COVID-19, and as such could be of use to other countries in the early stages of the disease outbreak.

Regarding your use of referencing, I found this to be adequate. However, as you will see in my specific comments on the attached annotated PDF, I felt there was a need to provide more context from other countries. The major selling point of this manuscript is why Taiwan appears to have been more successful at controlling COVID-19 than other countries, and without direct comparison I feel this might be a missed opportunity.

Regarding tables/figures, these were useful.

A major concern I do have however is your use of English which, whilst generally good, had several points for improvement. I have highlighted a number in the manuscript, but would recommend further detailed proofreading before re-submission.

Experimental design

This manuscript is in scope for the journal. As more of an epidemiological report, I found the methods generally sufficient to describe your approach, though I felt the lack of a clear timeline between case reports and introduction of measures to be confusing in places. I would recommend including a clear timeline of events to improve clarity. It would also be useful to include a scale for the measures introduced, as these more practical details would be invaluable to other countries considering implementing similar control measures.

I would further recommend the authors to carefully review whether various parts relating to 'normal' hospital practice in Taiwan would be necessarily known to those residing outside of Taiwan. I think there could be some value in quickly summarising what normal infection control measures, for example, would look like outside of the unique COVID-19 situation.

Validity of the findings

As a description of measures deployed, I found this manuscript to be sufficient. I further thank the authors for being careful to carefully acknowledge and qualify the fact that an exact causative link between lack of a significant outbreak and the control measures introduced could not be established. I would however recommend further comparison with other countries, as i stated earlier.

Reviewer 3 ·

Basic reporting

.

Experimental design

.

Validity of the findings

.

Additional comments

I think the paper is sound, the methods are well described. The results are well presented and are interesting for the reader.
However a minor revision is necessary at least and that would be my suggestion:
Some minor points: the language should be checked and corrected by a native speaker, there are many errors.
Fig.1 (Distrribution of cases and development in Asia is really not necessary, but in any case much to detailed,

---

## Round 0.2 · Minor Revisions

Still pending some modifications in your work. Please, see the reviewers' comments so as to have more information.

·

Basic reporting

Many thanks for sending this much improved manuscript for second review. It was a pleasure to read, was informative, and as such I think this has now very nearly reached publication standard. I do however have a small normal of minor modifications for consideration just to maximise the benefit of this paper, which I have described below.

Line 41: Consider using 'close contact' rather than physical, to also encompass droplet spread as a transmission mechanism.

Experimental design

I am happy with the presentation of this article - the addition of the timeline does greatly improve clarity, thank you. Just a couple of suggestions here:

Line 138-139: Regarding being moved to isolated rooms upon confirmation of COVID-19, it might be helpful to briefly outline, if possible what the stepdown procedure was for recovering cases i.e. what criteria were there to decide when patients could leave isolation. Deciding when patients can re-enter normal life is a significant area of debate, and one I feel you could contribute too.

Line 148: You have a few citations that are incorrectly formatted. Review these before publication.

Validity of the findings

As above, I am happy with your presentation and interpretation of findings. As such I have only a couple of suggestions:

Line 205: It might improve clarity by more carefully defining 'medical mask' and comparing it with more robust masks, such as N95. I'm guessing you are referring primarily to standard surgical masks here, but it might help to be a bit more specific, if possible.

Line 251: The herd immunity strategy in the UK has largely been abandoned. It might be good to acknowledge that for the reasons you stated why Taiwan did not follow this approach, the UK has now also changed approach... perhaps further justifying your decision.

Additional comments

Many thanks again for the opportunity to review this valuable manuscript. I noted in your rebuttal letter that you asked if I would be happy to be included in the acknowledgements section. Thank you so much for this thoughtful request, I am pleased that my reviews have helped you. I'm not sure if this journal publicly names reviewers anyway (in which case really don't worry), so have raised this with the editor.

Thank you also for the good wishes in the light of this pandemic - I hope you, your co-authors and families also continue to remain safe and well over the coming months.

Best wishes,
David

---

## Round 0.3 · accepted · Accept

All the reviewers' concerns have been correctly addressed.

Reviewer 1 ·

Basic reporting

Now, the manuscript is perfect.

Experimental design

Now, the manuscript is perfect.

Validity of the findings

Now, the manuscript is perfect.

Additional comments

The manuscript has been improved and looks great!

·

Basic reporting

No concerns - I would now consider this work publishable.

Experimental design

No concerns - I would now consider this work publishable.

Validity of the findings

No concerns - I would now consider this work publishable.

Additional comments

Many thanks for this further re-submission. The modifications you have made have further improved the article quality, and I particularly think the addition of your step-down strategy will prove useful. To date, although literature in this area is very fast moving, I haven't heard much discussion of criteria for removal of patients from isolation rooms. It is interesting to note that to my knowledge in my own country (the UK) patients are being moved to normal wards after a single negative test. In my opinion, considering the fairly poor sensitivity of this test, the UK strategy is madness. As such, any prompting of debate about a more cautious approach, such as undertaken by yourselves, is much needed and very useful. However, your acknowledgement of the time taken for patients to on average cross this boundary is also welcome - there are difficult compromises to be made, particularly as our understanding of the duration of infectious potential in patients remains poor. Anyway, I digress, I have recommended this article be published and look forward to seeing it in print! I wish you the best of luck for your continued control efforts, and share your hopes for a future less impacted by COVID-19.
Best wishes,
David